# Indicators of Attack Failure: Debugging and Improving Optimization of Adversarial Examples

## Abstract

Evaluating robustness of machine-learning models to adversarial examples is a challenging problem. Many defenses have been shown to provide a false sense of security by causing gradient-based attacks to fail, and they have been broken under more rigorous evaluations. Although guidelines and best practices have been suggested to improve current adversarial robustness evaluations, the lack of automatic testing and debugging tools makes it difficult to apply these recommendations in a systematic manner. In this work, we overcome these limitations by (i) defining a set of quantitative indicators which unveil common failures in the optimization of gradient-based attacks, and (ii) proposing specific mitigation strategies within a systematic evaluation protocol. Our extensive experimental analysis shows that the proposed indicators of failure can be used to visualize, debug and improve current adversarial robustness evaluations, providing a first concrete step towards automatizing and systematizing current adversarial robustness evaluations.

## 1 Introduction

Neural networks are now deployed in settings where it is important that they behave reliably and robustly [19, 15, 33, 3]. Unfortunately, these systems are vulnerable to *adversarial examples* [29, 4], i.e., inputs intentionally crafted to mislead machine-learning classifiers at test time. These attacks are especially important in settings where classifiers have security-critical consequences, including autonomous driving, automated medical diagnoses, and cybersecurity-related tasks such as spam and malware detection, web-page ranking and network protocol verification [27, 18, 26, 2, 28, 15].

This vulnerability has caused a strong reaction from the community, with many proposed defenses [33, 22, 31, 25]. Early defenses often argued robustness by showing the defense could prevent prior attacks, but not attacks tailored to that particular defense. As a result, most of these defenses have turned out to only provide a false sense of security, i.e., to be broken when targeted by an *adaptive attack* that tailors the attack strategy to the particular defense [11, 1]. More recent work has tried to evaluate using such adaptive attacks. Unfortunately, even this has proven difficult; recent work has shown that 13 published defenses proposed in the last year are ineffective despite almost all of them containing an analysis to adaptive attacks [30].

The reason why adversarial example defense evaluations are incomplete comes down to the difficulty of performing an adaptive attack, and diagnosing when they go wrong. Adversarial examples are typically generated through *gradient descent*: the adversary first constructs a *loss function* so that a minimum for that function is an adversarial example. While gradient-based attacks are highly effective at finding adversarial examples on undefended classifiers with smooth loss functions, many defenses substantially hinder the attack optimization by obfuscating gradients or by exhibiting harder-to-optimize loss functions. In particular, most attempted defenses to adversarial examples only succeed at increasing the difficulty of solving the minimization formulation, and *not* at actually increasing the

**Algorithm 1:** Our framework for computing adversarial attacks

---

**Input** : $\boldsymbol{x}$, the initial point; $y$, the true class of the initial point; $n$, the number of iterations; $\alpha$, the learning rate; $f$, the target model; $\Delta$, the considered region.

**Output** : $\boldsymbol{x}^{\star}$, the solution found by the algorithm

1 $\boldsymbol{x}_0 \leftarrow \mathtt{initialize}(\boldsymbol{x})$          ▷ Initialize starting point
2 $\hat{\boldsymbol{\theta}} \leftarrow \mathtt{approximation}(\boldsymbol{\theta})$      ▷ Approximate model parameters
3 $\boldsymbol{\delta}_0 \leftarrow \boldsymbol{0}$          ▷ Initial $\delta$
4 **for** $i \in [1, n]$ **do**
5    $\boldsymbol{\delta}' \leftarrow \boldsymbol{\delta}_i - \alpha \nabla_{\boldsymbol{x}_i} L(\boldsymbol{x}_0 + \boldsymbol{\delta}_i, y; \hat{\boldsymbol{\theta}})$      ▷ Compute optimizer step
6    $\boldsymbol{\delta}_{i+1} \leftarrow \mathtt{apply\text{-}constraints}(\boldsymbol{x}_0, \boldsymbol{\delta}', \Delta)$    ▷ Apply constraints (if needed)
7 $\boldsymbol{\delta}^{\star} \leftarrow \mathtt{best}(\boldsymbol{\delta}_0, ..., \boldsymbol{\delta}_n)$       ▷ Choose best perturbation
8 **return** $\boldsymbol{\delta}^{\star}$

---

robustness of the underlying classifier (i.e., increasing the actual distance of the decision boundary from the input sample) [10, 11, 1, 30]. Moreover, even though guidelines and best practices have been suggested to improve current adversarial robustness evaluations, the lack of automatic testing and debugging tools makes it difficult to apply these recommendations in a systematic manner. These difficulties have perpetuated a constant cat-and-mouse game where defenders propose new schemes, and attackers find that actually the defense was only increasing the difficulty of solving the underlying minimization problem [5, 3].

This paper directly addresses these limitations by (i) developing quantitative *indicators of failure*, i.e., metrics designed to help debug optimization of gradient-based attacks for generating adversarial examples, and (ii) suggesting a systematic evaluation protocol to improve current robustness evaluations by applying a sequence of specific mitigation strategies. In four case studies of published defenses that have been shown to be ineffective against stronger adaptive attacks, we show (i) that our indicators would have highlighted different failure modes in the original evaluations, and (ii) how these failures could have been easily overcome by following our suggested mitigation strategies.

**To summarize, we make the following contributions:** (i) we introduce a unified attack framework that captures the predominant styles of existing gradient-based attack methods, and allows us to categorize the five main causes of failure that may arise during their optimization (Sect. 2); (ii) we propose five *indicators of attack failures* (IoAF), i.e., metrics and principles that help understand why and when gradient-based attack algorithms fail (Sect. 3); (iii) we empirically evaluate the utility of our metrics on four recently-published defenses, showing how their robustness evaluations could have been improved by monitoring the IoAF values and following our evaluation protocol (Sect. 4; and (iv) we provide open-source code and data we used in this paper for reproducing resources. Our code is available at `https://github.com/ioaf-todo`.[1] We conclude by discussing related work (Sect. 5), along with the limitations of our work and future research directions (Sect. 6).

## 2   Adversarial Robustness: Gradient-based Attacks and Failures

We argue here that optimizing adversarial examples amounts to solving a multi-objective optimization:

$$\min_{\boldsymbol{\delta} \in \Delta} \left( L(\boldsymbol{x} + \boldsymbol{\delta}, y; \boldsymbol{\theta}), \|\boldsymbol{\delta}\|_p \right) , \tag{1}$$

where $\boldsymbol{x} \in [0, 1]^d$ is the input sample, $y \in \{1, \ldots, c\}$ is either its label (for untargeted attacks) or the label of the target class (for targeted attacks), and $\boldsymbol{\delta} \in \Delta$ is the perturbation optimized to have the perturbed sample $\boldsymbol{x}' = \boldsymbol{x} + \boldsymbol{\delta}$ misclassified as desired, within the given input domain. The target model is parameterized by $\boldsymbol{\theta}$. The given problem presents an inherent tradeoff: minimizing $L$ amounts to finding an adversarial example with large misclassification confidence and perturbation size, while minimizing $\|\boldsymbol{\delta}\|_p$ penalizes larger perturbations (in the given $\ell_p$ norm) at the expense of decreasing misclassification confidence.[2] Typically the attacker loss $L$ is defined as the Cross-Entropy (CE) loss, or the logit difference [11].

---

[1] Anonymized for submission.

[2] Note that the sign of $L$ may be adjusted internally in our formulation to properly account for both untargeted and targeted attacks.

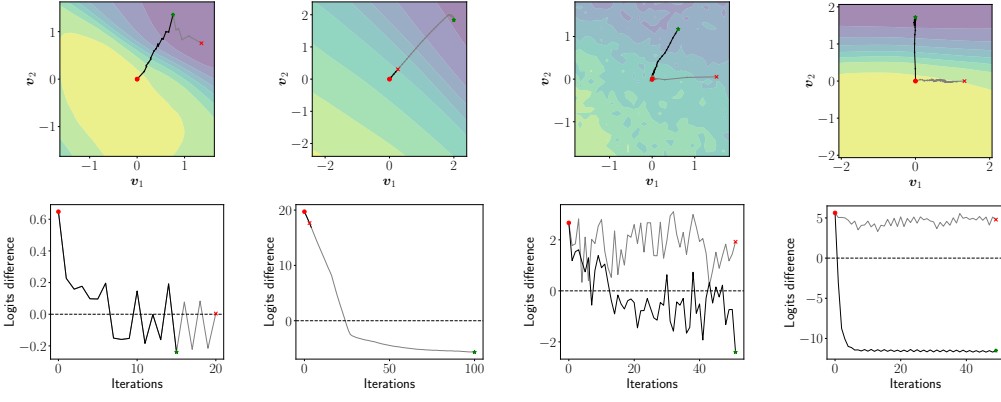

(a) Impl. problems.  (b) Non-converging attack  (c) Bad local optimum.  (d) Non-adaptive attack.

Figure 1: The four attack failures that can be encountered during the optimization of an attack. The failed attack path is shown in *gray*, while the successful attack is displayed in *black*. The point $\boldsymbol{x}_0$ is marked with the *red* dot, the returned point of the failed attack with a *red* cross, and the successful adversarial point with the *green* star. The top row shows the loss landscape, as $L(\boldsymbol{x} + a\boldsymbol{v}_1 + b\boldsymbol{v}_2, y_i; \boldsymbol{\theta})$. $\boldsymbol{v}_1$ is the normalized direction $(\boldsymbol{x}_n - \boldsymbol{x}_0)$, while $\boldsymbol{v}_2$ is a representative direction for the displayed case. In the second row we show the value of $L(\boldsymbol{x} + \boldsymbol{\delta}_i, y_i; \boldsymbol{\theta})$ for the evaluated model.

Multiobjective problems can be solved by establishing a different tradeoff between the given objectives along the Pareto frontier, by either using soft- or hard-constraint reformulations. For example, Carlini-Wagner (CW) [11] is a soft-constraint attack, which reformulates the aforementioned multiobjective problem as an unconstrained optimization: $\min_{\boldsymbol{\delta}} \|\boldsymbol{\delta}\|_p + c \cdot \min(L(\boldsymbol{x} + \boldsymbol{\delta}, y, \boldsymbol{\theta}), -\kappa)$, where the hyperparameters $\kappa$ and $c$ tune the trade-off between misclassification confidence and perturbation size. Hard-constraint reformulations instead aim to minimize one objective while constraining the other. They include maximum-confidence attacks like Projected Gradient Descent (PGD) [17], which is formulated as $\min_{\boldsymbol{\delta}} L(\boldsymbol{x} + \boldsymbol{\delta}, y; \boldsymbol{\theta})$ s.t. $\|\boldsymbol{\delta}\|_p \le \epsilon$, and minimum-norm attacks like Brendel-Bethge (BB) [6] and Decoupling-Direction-Norm (DDN) [24], which can be formulated as $\min_{\boldsymbol{\delta}} \|\boldsymbol{\delta}\|_p$ s.t. $L(\boldsymbol{x} + \boldsymbol{\delta}, y; \boldsymbol{\theta}) \le k$. In these cases, $\epsilon$ and $k$ upper bound the perturbation size and the misclassification confidence, respectively, thereby optimizing a different tradeoff between these two quantities.

The aforementioned attacks often need to use an approximation $\hat{\boldsymbol{\theta}}$ of the target model, since the latter may be either non-differentiable, or not sufficiently smooth [1], hindering the gradient-based attack optimization process. In this case, once the attacker loss has been optimized on the surrogate model $\hat{\boldsymbol{\theta}}$, the attack is considered successful if it evades the target model $\boldsymbol{\theta}$.

**Attack Algorithm.** According to the previous discussion, even if different attacks minimize different objectives or require different constraints, all of them can be seen as solutions to a common multiob-jective problem, based on gradient descent. Thus, their main steps can be summarized as detailed in Algorithm 1. First, an *initialization point* (line 1) needs to be set, and this can be achieved by directly using the input point $\boldsymbol{x}$, a randomly-perturbed version of it, or even a sample from the target class [6]. Then, if the target model $\boldsymbol{\theta}$ is difficult to deal with, or it is non-differentiable, the attacker must chose a surrogate model $\hat{\boldsymbol{\theta}}$ that approximates the real target $\boldsymbol{\theta}$ (line 2). The attack then iteratively updates the initial point searching for a better and better adversarial example (line 4), computing in each iteration one (or more) gradient descent steps (line 5) using the initial point and the perturbation $\boldsymbol{\delta}_i$ computed so far. Hence, the new perturbation $\boldsymbol{\delta}_{i+1}$ is obtained by enforcing the constraints defined in the problem (line 6), that can be updated accordingly to the chosen strategy [23, 24]. For *maximum confidence* approaches, the attack can not exit the $\Delta$ region, and samples are projected accordingly on this ball when reaching the constraints. Similarly, we consider *minimum distance* attacks successful only if they found adversarial examples inside the $\Delta$ region. At the end of the iterations, the attacker has collected all the perturbations along the iterations, formalized as the *attack path*. The final result of the algorithm is the the best perturbation contained in the attack path, w.r.t. the loss they are minimizing (line 7).

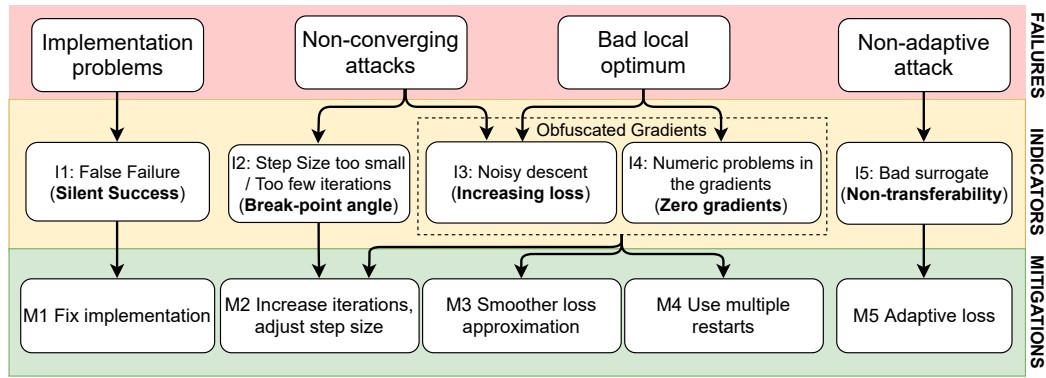

Figure 2: Indicators of Attack Failures. The top row lists the four general failures encountered in gradient-based attacks. The second row lists the Indicators of Attack failures we propose, and the last row depicts possible mitigations that can be applied.

## 2.1 Attack failures

We can now isolate four failures that can be encountered while optimizing adversarial attacks using Algorithm 1, and we bound each of them to specific steps of such procedure.

$F_1$: **Implementation Problems.** If no adversarial examples are found by the attack, it might be possible that the used implementation include errors or bugs. For example, we isolated a bug inside the procedure proposed by Madry et al. [17]. The attack as described returns the adversarial example only by looking at the *last* point of the attack path (line 7 of Algorithm 2), as shown in Fig. 1a, but would not return an adversarial example if one was found during search and then passed over.

$F_2$: **Non-converging attack.** When performing gradient descent based attacks, a common problem is that attacks do not converge to any local minimum, as shown in Fig. 1b. This problem can be caused by either the setup of the attack, and in Algorithm 1, this is reflected on the values of $\alpha$ and $n$, i.e. the step size of the attack, and the number of iterations. If $\alpha$ is too small, the gradient update step is not exploring the space (line 5 of Algorithm 1), while using too few iterations $n$ might cause an early stopping of the attack (line 4 of Algorithm 1). An example of this failure can be found in the evaluation of the defense proposed by Buckman et al. [7], where the authors only used 7 steps of PGD for testing the robustness of their defense, or by the one proposed by Pang et al. [21], where the defense has been evaluated with only 10 steps of PGD. Also, this failure might be triggered either by a too-large step size, that lead the optimizer to keep overshooting the local minimum, or the presence of *gradient obfuscation techniques* [31] that alter the gradients of the model to point to random directions, leading the descent to fail.

$F_3$: **Bad local optimum.** Once the attack reached convergence, the computed point might not be adversarial, since the optimizer has reached a region where it can not update anymore the adversarial perturbation, as shown in Fig. 1c. There are few reasons that might lead to such failure. One of them is again caused by the presence of gradient obfuscation, where the optimizer is unable to continue the descent, since it arrived in a region where the norms of gradients are (nearly) zero (i.e. flat regions), or again because the gradients are noisy, and the optimization lands on a bad local optimum (line 5 of Algorithm 1). An example of such failure is detected inside the defense proposed by Papernot et al. [22], where the model is trained to have signal in correspondence of samples, and producing regions with no gradient all around them. Another reason might be triggered by the choice of the initialization point itself (line 1 of Algorithm 1), that leads the optimizer into a region where no adversarial examples can be found. The latter has been detected by the analysis conducted by Tramèr et al. [30] against the defense proposed by Pang et al. [21], where a different initialization point lead the attack to find a better solution.

$F_4$: **Non-adaptive attack.** The loss function that the attacker optimizes does not match the actual loss of the target system, and this is caused by a bad choice of the surrogate model (line 2 of Algorithm 1), as shown in Fig. 1d. This issue manifests when either the attack is computed on an undefended model, and later tested against the defense, or the target model is not differentiable and the surrogate is not really approximating it. Since we consider both cases, we differ from the literature, where the

141 term *non-adaptive* has been used only for attacks that were not specifically designed to target a given
142 defense [30]. An examples of this failure is found in the defense proposed by Yu et al. [32], where the
143 attack has been computed against the undefended model, and then evaluated against the defense later.

144 To maximize the likelihood of creating successful attacks and hence avoiding such failures, current
145 recommendations [30] suggest to (i) select the strongest attacks against the model that is being tested;
146 (ii) state the precise threat model being considered; (iii) select the correct hyperparameters for the
147 attack being used; and (iv) compute charts to understand how the attacks behave by varying the size
148 of the perturbation. Indeed useful, such are only qualitative recommendations that require ad-hoc
149 inspection of each failed attack.

# 3 Indicators of Attack Failure

151 In this section we describe our Indicators of Attack Failures, i.e. tests that help an analyst debug a
152 failing attack. Each of these tests outputs a value bounded between 0 and 1, where values towards 1
153 implies the presence of the failure described by the test. Informed by the results of the indicators,
154 we propose potential mitigations that can resolve the presence of the detected failure. An overview
155 of such approach can be appreciated in Fig. 2, where we connect failures with the indicators that
156 quantify them, along with possible mitigations.

157 $I_1$**: Silent Success.** This indicator is designed as a binary flag that
158 triggers when the attack is failing, but a legitimate adversarial exam-
159 ple is found inside the attack path, as described by the implementa-
160 tion problem failure ($F_1$).

161 $I_2$**: Break-point angle.** This indicator is designed to quantify the
162 non-convergence of the attack ($F_2$) caused by the choice of too small
163 hyperparameters. We normalize the loss along the attack path and
164 the iteration, to fit the loss in the domain $[0, 1] \times [0, 1]$, and, ideally, a
165 well-converged loss should approximate a triangle in that domain, as
166 shown in Fig. 3. To create that triangle, we connect the first and the
167 last point in the loss curve, and we conclude the shape by considering
168 the point of the loss curve that is further to such conjunction. We are
169 interested in the amplitude of the basis $\beta$ angle, since it is the one

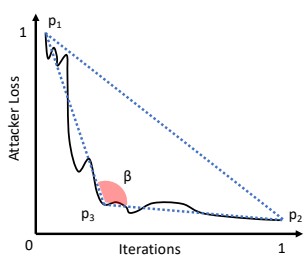

Figure 3: $I_2$ indicator.

170 that characterizes the shape of the triangle: when $\beta \approx \pi$, the triangle is flat, implying that the loss is
171 still decreasing. For this reason, the indicator computes $1 - |cos\beta|$, matching such intended behavior.
172 On the other hand, this indicator is close to 0 when the triangle is close to be right, hence $\beta \approx \frac{\pi}{2}$.

173 $I_3$**: Increasing loss.** This indicator is designed to quantify either the
174 non-convergence of the attack ($F_2$), or the inability of converging
175 to a good local optimum ($F_3$), both caused by the presence of noisy
176 gradients, where the loss of the attack is increasing while optimizing.
177 To characterize such behavior, we normalize the loss of the attack
178 and the iterations as we did in $I_2$, and we extract from it only the
179 portions where it increases, and we compute its area, as shown in
180 Fig. 4. When this indicator is close to 1, the values of the loss are
181 fluctuating around its maximum value, difficult to be decreased by
182 the optimizer.

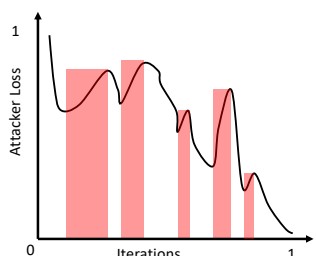

Figure 4: $I_3$ indicator.

183 $I_4$**: Zero gradients.** This indicator is designed to quantify the
184 bad-local optimum failure ($F_3$), caused by the absence of gradi-
185 ent information. For this reason, we compute how many times,
186 along the attack path, the gradients of the loss function are zero:
187 $\frac{1}{n+1} \sum_{i=0}^{n} \mathbb{1}_{\|\nabla_{x+\delta_i} L\|=0}$. This indicator is close to 1 when most of the norms of the gradient are 0,
188 causing the attack step to fail.

189 $I_5$**: Non-transferability.** This indicator is designed to quantify the non-adaptive failure ($F_4$), by
190 measuring if the optimized attack fails against the real target model, while succeeding against the
191 surrogate one. If the attack transfers successfully, the indicator is set to 0, otherwise it is set to 1.

### 3.1 Mitigate the Failures of Security Evaluations

Once the robust accuracy of a model has been computed, the attacker should now check the feedback of the indicators and mitigate accordingly the detected failures.

$M_1$: **Fix the implementation.** If $I_1$ is active, the attack is considered failed, but there exists an adversarial point inside the computed path that satisfies the attack objective. Hence, the resulting robust accuracy must be lowered to reflect this patch accordingly. Also, the attacker would want to run again their evaluations using another library, or a patched version of the same attack.

$M_2$: **Tune the hyperparameters.** If $I_2$ activates, it means that the optimization can be improved, and hence both the step size and iteration hyperparameters can be increased. Otherwise, if $I_3$ activates, the attack should consider a smaller step size, since the loss might be overshooting local minima.

$M_3$: **Use a different loss function.** If $I_3$ activates, and the decrement of the step size did not work, the attack should change the loss to be optimized [30], preferring one that has a smoother behavior. If $I_4$ activates, the attack should consider loss functions that do not saturate (e.g. avoid the softmax) [9], or also increase the step size of the attack to avoid regions with zero gradients.

$M_4$: **Consider different restarts for the attack.** If $I_3$ or $I_4$ activates, the attack might also consider to repeat the experiments with more initialization points and restarts, as the failure could be the result of added randomness or an unlucky initialization.

$M_5$: **Perform adaptive attacks.** Lastly, if none of the above applied, the attack might be optimizing against a bad surrogate model. If $I_5$ is active, the attack should be repeated by changing the surrogate to better approximate the target, or include the defense inside the attack itself [30]. This step implies repeating the evaluation, as the change of the surrogate might trigger other previously-fixed failures.

When attacks fail even after the application of recommended mitigations, it would be easy to assume that the evaluated defense is strong against adversarial attacks. However, the only thing known is that baseline attacks, properly tested, are not working against the defense. Hence, the designer of the defense should try as hard as possible to break the proposed defense with further investigations [12], and by performing sanity checks, e.g., ensuring that the robust accuracy drops to $0\%$ when the perturbation size is unbounded, or by trying different attack strategies, e.g., using gradient-free attacks or attacks designed by reversing the defense mechanism.

## 4 Experiments

We now exhibit the results of our experiments, by showing the correlation between the feedback of our indicators, and the false sense of security given by badly-evaluated defenses.

**Experimental setup.** We run our attacks on an Intel® Xeon® CPU E5-2670 v3, with 48 cores, 126 GB of RAM, and equipped with an Nvidia Quadro M6000 with 24 GB of memory. All the attacks and models have been wrapped and run by using the SecML library [20]. We select four defenses that have been reported as failing, and we show that our indicators would have detected such evaluation errors. For each of them, we set the hyperparameters for the attack as done in the original evaluation, in order to collect similar results.

*k-Winners-Take-All (kWTA)*, the defense proposed by Xiao et al. [31] uses only the top-k outputs from each layer, generating many discontinuities in the loss landscape, and hence resulting in the non-converging failure due to noisy gradients ($F_2$). We use the implementation provided by Tramèr et al. [30], trained on CIFAR10, and we test its robustness by attacking it with $\ell_\infty$-PGD [17] with a step size of $\alpha = 0.003$, maximum perturbation $\epsilon = 8/255$ and 50 iterations, with 5 restarts for each attack, scoring a robust accuracy of 58% on 100 samples.

*Distillation*, the defense proposed by Papernot et al. [22], works by training a model to have zero gradients around the training points, leading gradient-based attacks towards bad local optimum ($F_3$). We re-implemented such defense, by training a distilled classifier on the MNIST dataset to mimic the original evaluation. Then, we apply $\ell_\infty$-PGD [17], with step size $\alpha = 0.01$, maximum perturbation $\epsilon = 0.3$ for 50 iterations on 100 samples, resulting in a robust accuracy of 94,2%.

*Ensemble diversity*, the defense proposed by Pang et al. [21] is composed with different neural networks, trained with a regularizer that encourages diversity. We adopt the implementation provided by Tramèr et al. [30]. Then, following its original evaluation, we apply $\ell_\infty$-PGD [17], with step size $\alpha = 0.001$, maximum perturbation $\epsilon = 0.01$ for 10 iterations on 100 samples, resulting in a robust accuracy of 38%.

*Turning a Weakness into a Strenght (TWS)*, the defense proposed by Yu et al. [32], applies a mechanism

| Model | Attack | $I_1$ | $I_2$ | $I_3$ | $I_4$ | $I_5$ | $\bar{I}$ | RA |
|---|---|---|---|---|---|---|---|---|
| *k-WTA* [31] | PGD | 0.33 | 0.43 | 0.77 | - | - | 0.306 | 58,2% |
|  | APGD | - | 0.310 | 0.33 | - | - | 0.128 | 36,4% |
|  | PGD$^\star$ | 0.07 | 0.48 | 0.55 | - | - | 0.220 | 6,4% |
| *Distillation* [22] | PGD | - | 0.98 | - | 0.97 | - | 0.39 | 94.2% |
|  | APGD | - | 0.4 | 0.21 | - | - | 0.122 | 00.4% |
|  | PGD$^\star$ | - | 0.04 | - | - | - | 0.008 | 0% |
| *Ensemble Div.* [21] | PGD | - | 0.76 | - | - | - | 0.152 | 38% |
|  | APGD | - | 0.370 | 0.14 | - | - | 0.102 | 0% |
|  | PGD$^\star$ | 0.08 | 0.17 | 0.15 | - | - | 0.080 | 9 % |
| *TWS* [32] | PGD | - | 0.49 | 0.07 | - | 0.37 | 0.186 | 35% |
|  | APGD | - | 0.41 | 0.09 | - | - | 0.10 | 0% |
|  | PGD$^\star$ | - | 0.37 | 0.10 | - | - | 0.094 | 0% |

Table 1: Values of the Indicators of Attack Failures, computed for all the attacks against all the evaluated models. We denote the attacks that apply also the mitigations as PGD$^\star$.

for detecting the presence of adversarial examples on top of an undefended model, measuring how much the decision changes locally around a sample. Even if the authors also apply other rejection mechanisms, we take into account only the described one, as we wish to show that attacks optimized neglecting such term will trigger the non-adaptive attack failure ($F_4$). We apply this defended on a WideResNet model trained on CIFAR10, provided by RobustBench [14]. We attack this model with $\ell_\infty$-PGD [17], with step size $\alpha = 0.1$, maximum perturbation $\epsilon = 0.3$ for 50 iterations on 100 samples, and then we query the defended model with all the computed adversarial examples. While the attacks works against the standard model, some of them are rejected by the defense, resulting in a robust accuracy of 35%, highlighted by the trigger of the $I_5$ indicator. In this case, we consider an attack unsuccessful if the original sample is not misclassified and the adversarial point is either belonging to the same class, or it is labeled as rejected.

Each of these attacks have been executed with 5 random restarts. We also attack all these models with the version of AutoPGD (APGD) [13] that uses the difference of logit (DLR) as a loss to optimize. This strategy will take care to automatically tune its hyperparameters while optimizing, reducing possible errors that occur while deciding the values of step size, and iterations. Lastly, we compute attacks that take into account all the mitigations we prescribed, and they will be analyzed further on in the paper.

**Identifying failures.** We want now to understand if our indicators are correlated with faults of the security evaluations of defenses. We collect the results of all the attacks against the selected targets, and we compute our indicators, by listing their values in Table 1, along with their mean score. With a glance, it is possible to grasp that out hypothesis is right: the detection of a failure is linked with higher values for the robust accuracy, and also the opposite. Each original evaluation is characterized by high values of one or more indicator, while the opposite happens for stronger attacks. For instance, APGD automatically tunes its hyperparameter while optimizing, hence it is able to apply some mitigations directly during the attack. To gain a quantitative evaluation of out hypothesis, we compute both the p-value and the correlation between the average score of the indicators and the robust accuracy, depicting this result in Fig. 5. Both p-value and correlation suggest a strong connection between these analyzed quantities, confirming our initial belief.

**Mitigating failures.** We can now use our indicators to improve the quality of the security evaluations, and we apply the following pipeline: (i) we test the defense with a set of points with the original attack strategy proposed by the author of the defense; (ii) we select the failure cases and inspect the

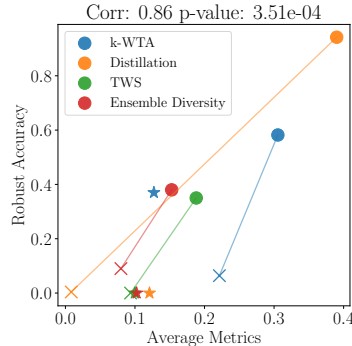

Figure 5: Evaluation of our metrics for different models. Robust accuracy vs. average value of the indicators, for the *initial evaluation* (denoted with '∘'), with the evaluation *after-mitigation* (denoted with '×'), and with APGD (denoted with '⋆')

| Model | Initial | $M_1$ | $M_2$ | $M_3$ | $M_4$ | $M_5$ | Final |
|---|---|---|---|---|---|---|---|
| *k-WTA* [31] | 58.2% | 36.4% | 36.4% | 6.4% | 6.4% | 6.4% | **6.4%** |
| *Distillation* [22] | 94.2% | 94.2% | 94.2% | 94.2% | 94.2% | 0.4% | **0.4%** |
| *Ensemble Diversity* [21] | 38.0% | 38.0% | 36.0% | 36.0% | 29.0% | 9.0% | **9.0%** |
| *TWS* [32] | 35.0% | 35.0% | 35.0% | 35.0% | 35.0% | 0.0% | **0.0%** |

Table 2: Robust accuracies (%) after patching the security evaluations with the prescribed mitigations.

feedback of our indicators *per-sample*; (iii) for each cause of failure, we apply the specific remediation suggested by the metric; and (iv) we show that the attack now succeeds, thus reducing the robust accuracy of the target model, and also the values of the indicators.

We report all the results of this process in Table 2, where each row shows the original robust accuracy, and how it is decreased, mitigation after mitigation. Also, all the individual values of each indicator computed on these patched attacks can be found in Table 1, marked as PGD$^\star$.

*Mitigating k-WTA failures.* For many failing attacks, the *I1* indicator triggers, implying that the attack found an adversarial example inside the path. We then apply mitigation $M_1$, and we lower accordingly the robust accuracy of the model to 36,4%. We then analyze the feedback of the $I_3$ indicator, the one that detects the presence of noisy gradients. We apply mitigation $M_3$, and we change the loss of the attack as described by Tramèr et al. [30]. This loss is computed by averaging the gradient of each single point of the attack path with the information of the surrounding ones. The resulting direction is then able to correctly descent toward a minimum. We run $\ell_\infty$-PGD with the same parameters, but smoothing the gradients by averaging 100 neighboring points from a normal distribution $\mathcal{N}(\mu = \boldsymbol{x}_i, \sigma = 0.031)$, where $x_i$ is a point in the attack path. After such mitigation, the robust accuracy drops to $6, 4\%$, and so follows the indicator (Fig. 6a).

*Mitigating Distillation failures.* All the attacks fail because of the absence of gradient information, leading the attack to a bad local optimum ($F_3$), and such is highlighted by the feedback of the $I_3$ indicator. We apply mitigation $M_3$, and we change the loss optimized during the attack, following the strategy applied by Carlini et al. [9], that computes the loss of the attack on the logit of the model rather than the final softmax layer. We repeat the PGD attack with such fix, and the robust accuracy drops to 0%, along with the indicator $I_3$ (Fig. 6b).

*Mitigating Ensemble diversity failures.* Firstly, the $I_1$ indicator highlighted the presence of $F_1$, implying that some failing attacks are due to the implementation itself. We apply mitigation $M_1$, and the robust accuracy decreases to 36%. Also, $I_2$ indicator is active, implying that the loss of of failing attacks could be optimized more. For this reason, we apply mitigation $M_2$, and we increase the step size to $0.05$ and the iterations to $50$. This patch is enough for lowering the robust accuracy to $9\%$. (Fig. 6c).

*Mitigating TWS failures.* The detector is rejecting adversarial attacks successfully computed on the undefended model, triggering the $I_5$ indicator. Hence we apply mitigation $M_5$, and we adapt the attack to consider also the rejection class. This version of PGD minimizes the usual loss function of the attacker, but it also minimizes the score of the rejection class when encountered, allowing it to evade the rejection. We run such attack, and we obtain a new robust accuracy of 0% (Fig. 6d).

# 5 Related Work

**Other systematic analysis on robustness evaluations.** There have been a number of prior papers evaluating the robustness of particular defense schemes [10, 1, 30]. These papers focus on understanding whether the robustness claims of particular defenses are true, often by performing one-off attacks or by proposing new general attack approaches that can be used to break future defenses. In contrast our goal is not to break any particular defense, but rather to help researchers understand when their evaluation may have gone wrong. In this way our paper is related to Carlini *et al.* [12] that systematizes various suggestions from the literature for how to ensure that adversarial robustness evaluations are performed thoroughly. We imagine that our tests could be included in future recommendations for robustness evaluations.

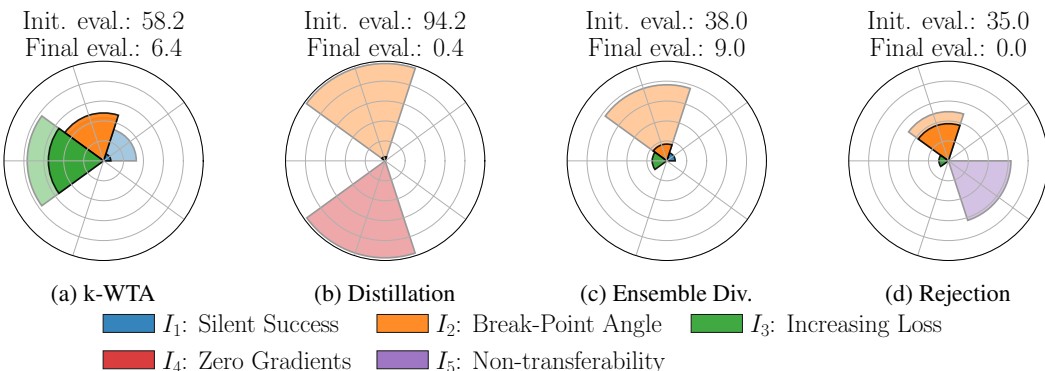

Figure 6: The values of our indicators and the success rate (SR) of the attack, before (semi-transparent colored area) and after (solid colored area) fixing the failures, computed for the analyzed models.

**Benchmarks.** Related to this work, there are a number of attack benchmarks that have been constructed. Instead of measuring the robustness of individual schemes as the prior papers do, these benchmarks aim to provide a complete evaluation framework that can be applied to any future defense as well. Ling et. al [16] proposed DEEPSEC, a benchmark that tests several attacks against a wide range of defenses. However, this framework was shown to be flawed by several implementation issues and problems in the configuration of the attacks [8]. Croce et al. [14] propose RobustBench [14], that accepts state-of-the-art models as submissions, and it tests their robust accuracy by applying AutoAttack [13]. However, this benchmark suite only works on CIFAR-trained models, and it is not able to determine which are the possible causes of such scored performance.

Hence, these benchmark would benefit from our indicators, since they might provide useful insight that can be autonomously computed. Here we imagine that our framework could be used to help these tools automatically detect when their evaluations are incomplete, so that they could warn the operator that there was a potential error that should be investigated.

## 6    Contributions, Limitations and Future Work

We propose the Indicators of Attack Failures (IoAF), quantitative tests that help the debugging of faulty-conducted security evaluations, and we propose a pipeline for mitigating their issues, leading to a fairer evaluation. We select defenses that have been previously shown to be weak against adversarial attacks, and we evaluate them with the lens of our indicators, showing that we could have detected their misconduct in advance. We empirically prove that these test are correlated with wrongly high robust accuracy, while they drop when attacks are successful.

On top of these contributions, we acknowledge some limitations in our methodology. We do not provide a fully-autonomous way for deciding how to turn an attack into its adaptive version against a particular defense (e.g. gradient obfuscation), but we provide quantitative tools for helping the decision among all the possible solutions that the attacker could come up with. Another limitation lurks in the choice of the attack itself, since some unknown-and-adaptive attack could behave very differently w.r.t. standard one, triggering some indicator in the process. However, these tests can be patched accordingly to take care of these newly-proposed patched attacks, and still being used as debugging tools. Lastly, as already discussed in Sect. 3, if the evaluated defense is not triggering any indicators it does not imply it is secure, but rather it forces the application of other sanity checks [12]. We believe some part of this last process can be automatized with additional indicators, however we leave this as a future work.

We hope that future work will include our indicators during the evaluation phase of new methods, in order to identify when attacks are failing for known reasons, and thus contributing to the creation of better defense mechanisms. Also, this work pose a preliminary step towards the creation of interactive dashboards that can be inspected as a web application. Finally, it would be insightful to attach our pipeline of indicators and mitigations to already-available benchmarks (i.e. RobustBench [14]), possibly detecting other failures in security evaluations we did not covered in our experiments.

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
