# OpenReview forum: "Indicators of Attack Failure: Debugging and Improving Optimization of Adversarial Examples"
_NeurIPS.cc/2021/Conference — NeurIPS 2021 Submitted_

### Official Review · Reviewer_9wg3 · 2021-06-27

**Rating:** 3
**Confidence:** 4

**Summary:**

The paper, similar to [1], proposes a systematic way to debug problems that may arise when implementing adversarial attacks to evaluate a proposed defense. They categorize the implementation problems into 4 categories, propose 5 indicator metrics that can identify what problem exists in an implementation, and highlight 5 mitigation strategies that can be used when a particular indicator is triggered. In the context of 4 existing defenses, they show that original Projected Gradient Descent attack implementation overestimates the security of these defenses. By leveraging the proposed indicator values and using their suggested mitigations to properly generate PGD adversarial examples, they are able to show that the attack is much more lethal and can invalidate the existing defenses. They compare the efficacy of their manual approach for debugging with AutoPGD, which tunes hyperparameters automatically to improve the PGD attacks chances of success.

[1] Carlini, N., Athalye, A., Papernot, N., Brendel, W., Rauber, J., Tsipras, D., ... & Kurakin, A. (2019). On evaluating adversarial robustness. arXiv preprint arXiv:1902.06705.

**Limitations And Societal Impact:**

See main review.

**Main Review:**

### Strengths

- The paper is easy to follow.
  1. Provides an abstract view of adversarial attacks and shows where implementation downfalls arise,
  2. Proposes a set of indicators to identify these downfalls,
  3. Suggests which existing mitigation strategies to leverage when a indicator triggers,
  4. Shows the efficacy of this manual process to debug attack implemenation by comparing it to automatic approaches.

- The paper treats a subset of the issues highlighted in [1] in a quantitative fashion by proposing indicator functions.


### Weaknesses / Correctness / Clarification

- The main novelty of the paper seems to be proposing the indicator functions. Note that the universal skeleton of attack algorithms and the mitigation strategies are already known. Perhaps, tying them together in the context of the indicator functions (i.e. parts of the algorithm that trigger it and mitigations that need to be used) is another novelty.

- On the indicator functions, I have a few concerns, clarifications and perspectives. As an overall comment, I suggest you spend more time describing the challenges you faced to come up with these metrics, describe them with more rigor and highlight how it can take care of edge scenarios (Many of the figures/tables have redundant information; hence, space should not be a problem). More specifically,
  - I_1 (silent success) and I_5 (non-transferability) are zeroth-order strategies that are often used by researchers who try to debug the efficacy of an attack algorithm. While these simple measures do show that not all researchers did due diligence when evaluating defenses, I am not convinced that these indicators address any challenge-- it is good software development guidance.
  - For I_2 (break-point angle), the authors state "we conclude the shape by considering the point of the loss curve that is further to such conjunction" (lines 168-169). I unfortunately did not understand this. I wanted to evaluate my understanding on some cooked up complicated loss plots but couldn't as I wasn't sure how to concretely identify this third vertex from the statement above.
  - For I_3 (increasing loss), is there some sort of formal way to say that this measure of area will always be (an admissible) heuristic for evaluating if the attack can converge to a good local optima? While for figure 4 it does make sense, I am not sure if it is always going to work. Even a proof for a weaker statement like _Given two attacker loss plots (for the same model, data blah..), if one converges to a better local optima than the other, the value of I_3 will always be smaller for it_ should work.
  - For I_4 (zero gradients), again some sort of formal treatment + highlight what was the major challenge one needed to solve to come up with it would be helpful.

- From table 1, I see that APGD has higher values for several indicator functions, such as Distillation I_2 (0.4), Ens. div. I_2, TWS I_2 then the mitigated PGD*. Yet, in all 3 cases, Auto PDG is almost similar or better than PGD*. This makes me question the validity of I_2 as a good indicator. Similarly for I_3, APGD has almost the same value as Ensemble Div for APGD (0.14) and PGD* (0.15) and yet APGD is much more effective at mitigating implementation issues. As stated above, without some formal guarantees, it is hard to accept them as indicators on face value.

- Evaluation against AutoPGD shows that this manual effort is better only in the context of debugging attack implementation for k-WTA. For others, its almost it's either a bit worse (ensemble div) or similar (distillation, TWS). At this point, I am not convinced why should one rely on manual debugging techniques (which may have pitfalls of their own).

- As all gradient based attacks (eg. CW, PGD, BB, DDN) can be represented as part of their framework, wondering why evaluation only looks at PGD? They should create an auto-version of the other attacks and adapt them as baselines to compare against. Simply showing efficacy of indicators based on only one attack type (i.e. PGD) isn't enough.

- The indicators provide a user confidence that many obvious issues do not exist but is not sufficient to say that having mitigated all the issues with the attack implementation, if the defense is really resilient (217-219).

- Table 1 shows a subset of indicators are used for each defense-attack combo, but Table 2 (sort of) implies all mitigations are applied in a step-by-step fashion. Not sure if a particular sub-set of indicators are triggered, why are all mitigations applied? If not, please ensure the tables don't mislead the reader.

- The (pretty) figures (Fig 5 and Fig 6) seem redundant. All of this info is already there in the tables (or am I missing some critical insight here?).

#### Minor

- (245) Streng`ht` -> Streng`th`
- (Table 1) Replace `,` with `.` in accuracy values 58`,`2% -> 58`.`2%
- (249) defen`ded` -> defen`ces`
- (301) 6`,`4% -> 6`.`4%


[1] Carlini, N., Athalye, A., Papernot, N., Brendel, W., Rauber, J., Tsipras, D., ... & Kurakin, A. (2019). On evaluating adversarial robustness. arXiv preprint arXiv:1902.06705.

**Time Spent Reviewing:**

6

---

> ### Author Response · Authors · 2021-08-09
> **Reply to Reviewer 9wg3**
>
> We thank the reviewer for the in-depth discussion they brought for our work, and we take the chance to clarify and better explain some key points of such discussion.
>
> Before delving into point-to-point responses, it is worth remarking that **indicators are tools for debugging and not for comparing attacks and defenses**. The purpose of the indicators is to detect and avoid common failures. If an evaluation does not trigger any indicator, it does not imply that the model is robust. It only means that the evaluation is not affected by any known failure for which an indicator is available. However, there may be unknown causes of failure that undermine the current evaluation, but we’re just not aware of them. The debugging role of indicators is exactly similar to that of software testing; if you don’t find any bugs or security vulnerabilities with testing, it doesn’t mean that your software doesn’t have any. In this sense, the indicators should not be used as a tool for comparing attacks or defenses but rather as a tool that reveals the false sense of security provided by a flawed evaluation. Similarly, indicators have to be developed to characterize known causes of failures and should not be expected to be able to identify new causes of failures or show how to break new defenses.
>
> **Concerns on indicator functions.** We now discuss all the issues raised around the choice of our indicators.
>
> * **Clarification on I1 and I5.** While it is true that both I1 and I5 are zeroth-order and trivial strategies that developers can apply, we find them missing in most of the state-of-the-art evaluations [1]. To mention one example, with indicator I1, we detected an issue in the PGD version proposed by Madry et al. that is used in many adversarial libraries, including FoolBox and CleverHans, which are two of the most used libraries for adversarial robustness evaluation. Without this fix, they might return a suboptimal point, which is not adversarial, although an adversarial example was found during the optimization procedure.
> Indicator I5 is trivial too, but many defense papers do not consider this easy check [1], and therefore they are broken suddenly after being published.
> We believe that performing those simple checks is crucial to reduce the number of published defenses incorrectly evaluated.
> * **Clarification on I2.**  Indicator I2 checks if the loss stabilizes to a certain value. To this end, we select the values at the first and last steps, as they give an indication of the trend in the loss, and we add a further point between them (but still on the loss curve) to characterize the shape of the curve. This third point is the one on the loss curve that is further away (with the largest perpendicular distance) from the segment connecting the first and last points on the same curve. Using these 3 points, we build a triangle, and we measure the angle at its base. If the angle is small (near 90°), it means that the loss reached a minimum in an iteration and during the following iterations is not substantially improving (i.e., it has converged). If such an angle is instead wider (e.g., almost 180°), it means that the loss curve is almost a straight line; hence it is decreasing, and it may further decrease by performing more iterations. We apologize if the explanation was not clear, and we will clarify this aspect in the paper.
> * **Clarification on I3.** Unfortunately, no formal methods can state if a point is the global minimum or maximum of a non-convex/non-linear function; hence we rely on the fact that, if the loss is increasing over the iterations, the final solution is very likely not to be a good local optimum. Also, since this indicator characterizes the path followed while optimizing, which is different from attack to attack; hence the value of this indicator is not comparable across different attacks.
> * **Clarification on I4.** We check if the product of the gradient and step size (line 5 of Algorithm 1) is zero by checking the norm of the computed gradient (ignoring the trivial case where the step size is zero). This encompasses both the presence of distillation and flat regions.
>
> **On the similarity of performances of APGD and PGD\*.** For some cases, we had to consider APGD versions that adapt to the specific case, and we had to apply mitigations. For example, for distillation, we had to use M3, taking a version of the attack that does not use the Cross-Entropy Loss as the objective function. Otherwise, it would have suffered from zero gradients just as standard PGD. For TWS, we had to consider multiple repetitions in order to achieve the robust accuracy that we provide in the table, hence using M4. Overall, this shows that our IoAF can also help debug and improve adversarial robustness evaluations carried out with APGD.
>
> **On the effectiveness of debugging with the indicators.** AutoPGD has shown good performances on the majority of the considered defenses.
> However, there is no guarantee that APGD won’t be affected by some of the failures as well. In our analysis (and also noticed by the reviewer), that is exactly the case of kWTA, which performs gradient obfuscation, where automatic hyperparameter tuning of APGD did not help in finding the adversarial examples. In this case, if the designer wouldn’t have applied our debugging indicators, they wouldn’t have found as many adversarial examples, reporting an overly-optimistic robustness evaluation.
> Moreover, as explained in [1], newly-proposed defenses often require devising adaptive attacks, which may be subject to failures, and, therefore, their evaluation would be easier by employing the proposed indicators of failure.
>
> **On the choice of using only PGD.** Due to space constraints, we chose PGD as the sole case study since it is the most used attack for evaluating defenses. Our analysis may be extended to more attacks in a future, extended version of this work.
>
> **On the interpretation of non-detected failures.** In the paper, we are not stating that a defense is robust if no indicator triggers (213-215), but only that the evaluation is not affected by any known failure.
>
> **On the application of mitigations.** We specify in the paper that the process is sequential and devised to be applied in a per-sample fashion (line 283-288). We'd also like to remark that future developments of this work may try to leverage indicators for automatically improving the hyper-parameter tuning of attack algorithms.
>
> **Clarification on figures 5 and 6.** Both figures have been added to better highlight aspects of our work that might be difficult to grasp from tables. Figure 5 shows the correlation between the average metrics and flawed evaluations reporting high robust accuracy. When the evaluations are patched, the indicators and the robust accuracy decrease. Figure 6 shows that the mitigations reduce (on average) the values of the corresponding indicator.
>
> We hope that these considerations would be helpful to better understand the scope and the efficacy of our work, and we hope that these insights might help the reviewer to reconsider their final judgment of the paper.
>
> [1] Florian Tramer, Nicholas Carlini, Wieland Brendel, & Aleksander Madry. (2020). On Adaptive Attacks to Adversarial Example Defenses.

---

> > ### Comment · Reviewer_9wg3 · 2021-08-31
> > **Thanks for the response; unfortunately, I am still not convinced.**
> >
> > The authors show due diligence in the rebuttal and I would like to thank them for it. However, my key concerns remain.
> >
> > - The indicator functions and mitigations are either already known (also pointed out by Reviewer ZPse) or lack sufficient contribution (trivial or devoid of any formal, even weak, guarantees).
> > - The authors repeatedly specify their approach is only meant to serve a debugging process that provides more confidence in the implementation of attack algorithms. Coupled with lack of novelty for the machine learning community (previous point) makes this an appropriate software manual / a documentation / a best-practice guide (and may be appropriate for software/systems/security conferences); I am not convinced about its relevance to the NeurIPS audience.
> > - Evaluation with only PGD due to space constraints (rebuttal point "On the choice of using only PGD") + No good reason to eat up extra space for Figure 5 and 6 (rebuttal point "Clarification on figures 5 and 6" doesn't deny my review that the figures have the same information already present in the tables) seems weird. Without empirical evaluation on other attacks (especially given the main claim of the paper is to sever as a debugging tool in empirical settings) makes it difficult to justify the effectiveness of their work (note that no formal analysis anyway exists).

---

> > > ### Author Response · Authors · 2021-09-06
> > > **Response**
> > >
> > > We thank the reviewer for the clarifications, but substantially disagree on many points. We thus want to remark once again that this work has been the first to define and propose the use of quantitative indicators to detect failures in the evaluation of adversarial robustness via gradient-based attacks (as also pointed out by reviewer wRYi, Ypvj and ZPse). Such quantitative measures of failure have neither been envisioned nor used before. Moreover, while the mitigations we have considered in our work are known, our work suggests how to apply them according to a systematic protocol, depending on the set of indicators of failure triggered by the evaluation (rather than being limited to provide a set of generic, recommended evaluation guidelines). We do believe that this aspect can also be considered novel with respect to the state of the art. To further support our arguments, we would finally like to point out that, even in the presence of already-known guidelines/mitigations, researchers have been committing the same mistakes over and over, as reported in the following examples:
> > >
> > > * all the papers described by Tramér et al. (https://arxiv.org/abs/2002.08347) were proven broken after the evaluation guidelines had already been known and published, i.e., they were broken using known techniques;
> > > * DeepSec (https://ieeexplore.ieee.org/document/8835375) was highly criticized after its acceptance (https://github.com/kleincup/DEEPSEC/issues), despite the related evaluation guidelines had already been developed and published;
> > > * Foolbox itself has implementation issues related to the fact that PGD iterates using unit-norm gradients with fixed step size, and it is thus not guaranteed to converge to the optimal solution.
> > >
> > > To conclude, we disagree with the reviewer on the criticisms about the novelty of our work. We will anyway do our best to extend our work by improving our case studies and experimental analysis.

---

### Official Review · Reviewer_wRYi · 2021-07-15

**Rating:** 8
**Confidence:** 4

**Summary:**

This paper reasons about complications that may cause authors to not recognized that their newly proposed defense is ineffective. The authors identify failure motivations, propose 5 indicators (measurable) that are indicative of an attack failure, and also propose appropriate mitigations. The paper very methodically and experimentally presents this findings, and offers a more concrete support towards adaptive attacks evaluations whenever new defenses are proposed.

**Limitations And Societal Impact:**

Mostly comments regarding the interpretation of the average scores, and how the indicators may be useful for further adaptive attacks.

**Main Review:**

# Strengths
- Very important line of research, focused on supporting and understanding how to systematically carry out adaptive attacks.
- Contributions, significance and novelty is clear.
- Systematic evaluation of indicators of attack failures in existing defense robustness evaluations
- Code scheduled for release.

# Weaknesses
- Mitigation $M_5$ seems a bit vague in terms of actionable points.
- The code  link seems broken. Not sure if it was just meant as a placeholder for future publication of code?
- "Robust accuracy" has not been formally defined
- Unclear interpretation of a 'good' or 'bad' value for the indicators fo attack failures (e.g., Figure 1)

# Detailed Comments

The indicators presented are derived from very nice intuitions and quantify effects.  I think this is a very systematic and well-thought work that can improve the state of the art in terms of best practices when evaluating defenses. I do have some comments that could improve the quality of the work.

## Mitigations

$M_1$ through $M_4$ are quite clear. $M_5$ starts to be a bit fuzzy, but the last paragraph of Section 3 should require more discussion. If I understand correctly, you are suggesting that the whole framework you propose for indicators of attack failure is just for the following:

> the only thing known is that baseline attacks, properly tested, are not working against the defense

What can be the utility of the indicators of attack failures if you try to propose a more advanced adaptive attack upon further investigation?

## Experimental evaluation

The authors conduct a thorough experimental evaluation considering also many existing defenses. However, there are some things that should be better specified:
- **Definitions**. There is no formal definition of "Robust accuracy".
- **Parameters**. You perform 5 random restarts of the attacks. Is this a number you recommend also future people to use, or is it a good number as long as there are no indicators of attack failures that activate?
- **Interpretation of the indicators**. In Figure 5, you show the average score of the indicators on the X-axis. What I see is for example that the average scores for k-WTA are higher both before and after mitigations than most of the other scenarios (e.g., TWS and Ensemble Diversity). Moreover, it seems you have an average still >0.2 for k-WTA. How should readers interpret such results, and know whether the average score of the indicators is sufficiently low?

## General comment

I may be playing the devil's advocate here, but it would be interesting to have the following discussion:
- Do you feel that an attacker may take advantage of the fact that you proposed the indicators of attacks failure?
In a sense, can an attacker take advantage of the fact that a defender is using indicators of attack failure to evaluate the robustness of their defense? Maybe an attacker that proposes an apparently robust algorithm but of which he knows some other weaknesses?

## Minor comments

- Figure (1) may also use a different line-style because black and gray look a bit similar to my eyes. Maybe also a different line width? Or a greater color difference?

**Time Spent Reviewing:**

5

---

> ### Author Response · Authors · 2021-08-09
> **Reply to Reviewer wRYi**
>
> We thank the reviewer for the very detailed review of our work, and for having captured its core contributions. We now take the chance to address the main weaknesses expressed by the reviewer.
>
> **Mitigations.** We agree with the reviewer that M5 may appear a bit vague. The reason is that an adaptive attack needs to be tailored to the targeted defense, which may include never-before-seen defense strategies. It is thus clear that, even if some general advice exists, there can not be any systematic approach to devising adaptive attacks.
> We will however expand the corresponding discussion to include the case in which external detectors are used, for which one may adjust the loss function by adding a penalty term to ensure that the attack also tries to bypass the detection mechanism. In any case, the utility of the indicators of attack failure here is that they actually allow us to know whether a novel, adaptive attack should be developed, or if patching known attacks may already suffice to provide a reliable adversarial robustness evaluation.
>
> **Code link is broken.** The URL of the code we wrote in the paper is only a placeholder for preserving the double-blindness of the submission; we will substitute it with the actual URL in case of acceptance. The code was included within the supplementary material.
>
> **Experimental evaluation.**
>
> * **Definitions.** Robust accuracy is defined as the accuracy of the classifier measured on the perturbed (adversarial) examples. We will clarify this point in the paper.
>
> * **Parameters.** We used 5 restarts in our experiments as they were sufficient to improve the defense evaluations, as also witnessed by the indicators’ value. More restarts can be normally used at the expense of a higher computational complexity. We will clarify this aspect in the paper.
> * **Interpretation of Indicators.** As we have explained in our main response and in response to reviewer ZPse too, the proposed indicators are devised to detect known attack failures. Figure 5 is only meant to show that there is a correlation between the average metrics and flawed evaluations reporting high robust accuracy. Unfortunately, the values of the indicators depend on the behavior of the loss function across iterations, so they depend on the optimization process followed by the attack and on the target model; accordingly, one may just want to ensure that, after applying the corresponding mitigations, their values do not decrease any further. For k-WTA, one may probably consider increasing the number of attack restarts to 1,000 as in [30], to further smooth the optimization process and potentially decrease the value of the corresponding indicator. However, we found that even using just 5 restarts already has a significant impact on decreasing its robust accuracy.
>
> **General Comment.** The reviewer poses an interesting question for future research. Indeed, there might be ways to trick the proposed indicator of failures, but this would probably require additional effort to the attacker, and considering different threat models. The indicators may also be used to further automate the development of improved attacks/evaluations, by systematically applying the suggested mitigations.
>
> We hope to have answered the reviewer’s curiosities, and that we shed some light on the points that resulted unclear.

---

> > ### Comment · Reviewer_wRYi · 2021-08-31
> > **Feedback to authors**
> >
> > Thank for your detailed answer. I genuinely thought I replied here earlier, but while I was going through all reviews again, I noticed I did not. I wanted to take time to acknowledge that I read the authors response, and I maintained my positive opinion on the paper. I believe this is a very important work that could benefit several communities relying on ML, especially in systematically reviewing common mistakes that lead to overestimation of defense effectiveness.

---

> > > ### Author Response · Authors · 2021-09-01
> > > **Acknowledgement**
> > >
> > > Thanks for appreciating the novelty of our work and for taking the time to respond. Our goal was indeed to provide a methodology towards systematizing and improving robustness evaluations, avoiding overly-optimistic evaluations and the common mistakes which have been made over and over by the community. We really appreciate that the reviewer definitely understands the goal of our work, and the relevance of the problem it addresses.

---

### Official Review · Reviewer_Ypvj · 2021-07-16

**Rating:** 6
**Confidence:** 4

**Summary:**

This paper proposes a systematic method for understanding the failures of adversarial attacks as means towards thorough evaluation of robustness defenses in the research community. The devised framework relies of three connected aspects: (i) analysis of the main causes why gradient-based adversarial attacks fail, (ii) metrics for identifying their reason of failure and (iii) practical mitigations to ensure attack efficiency when evaluating defenses. All these aspects are illustrated and confirmed on four existing defenses that were initially subjected to limited evaluation.

**Limitations And Societal Impact:**

Yes

**Main Review:**

Clarity
- Very clear and well-organized paper.
- The code for reproducing the experiments was provided.

Originality:
- The paper formalizes what is common knowledge in the community. However, I do appreciate the effort that goes into this formalism and the necessity for the community to have such a methodology to rely upon.

Quality and soundness:
- The motivation of the paper is clear, and I agree that, despite existing guidelines for robustness evaluation, robustness benchmarks in published research are often incomplete or misleading.
- This work is in line with previous guidelines on robustness evaluation.
- Previous work in the fields of robustness evaluation, broken defenses and machine learning security evaluation platforms are cited appropriately.
- The proposed analysis of four existing defenses is thorough and consistent with the proposed evaluation methodology.

Soundness and significance:
- While I appreciate the proposed framework and its application to the four chosen defenses, a more extensive experimental study would probably be more convincing, as the devised framework is mostly based on observations (as opposed to a formal proof of any kind). There is an abundance of published defenses with incomplete evaluation. Showing the applicability of the contribution to a wider range of defenses would give it better odds of reaching its target: becoming part of the de facto robustness evaluation standard. This suggestion is in line with previous publications proving the lack of robustness in prior methods (e.g., [1, 10] in the paper), where around ten defenses are examined in each.

Minor:
- Fig. 1b: colours seems to be mixed-up between the successful (shown in grey, but should be black) and failed attacks.
- Tab. 1: the explanation for the column name RA should be introduced in the table caption or the paragraph commenting the table.

**Time Spent Reviewing:**

3h

---

> ### Author Response · Authors · 2021-08-09
> **Reply to Reviewer Ypvj**
>
> We thank the reviewer for appreciating our work, and for their comments on the experimental evaluation. We want to clarify our intent in the choices of the defenses. We agree that adding more evaluations would be beneficial for demonstrating the indicators of failures in action; however, due to space constraints, we have selected some interesting cases covering different defense mechanisms, which we think are well representative of the different failures captured by the indicators. As we have previously explained in our main response and in response to reviewer ZPse too, we have considered already-broken defenses to show that the proposed indicators can correctly predict known, flawed evaluations, and hopefully help prevent them in the future.

---

> > ### Comment · Reviewer_Ypvj · 2021-09-01
> > **Thank you for your response**
> >
> > I would like to thank the authors for their response to all the reviewers. It is clear to me that the topic of the paper is important and that the community could use more formalization around best practices.
> >
> > Unfortunately, my main concerns still stand. To be reasonably convincing, the proposed indicators would either have to be theoretically founded or more widely evaluated. This issue has been raised by multiple reviewers, including suggestions for covering more defenses and other attacks. Addressing these limitations would most likely establish the proposed indicators as part of the good practices in the community. As to space constraints, the main paper could include a summary of an extensive evaluation of the indicators and some detailed examples on a few defenses / attacks. Everything else could find its place in the Appendix. With the inclusion of the source code (as currently done), the evaluation would remain reproducible.

---

> > > ### Author Response · Authors · 2021-09-06
> > > **Acknowledgement**
> > >
> > > We thank the reviewer for this frank comment, appreciated. We will work towards improving our methodology and extending our experimental analysis to make it more sound and convincing.

---

### Official Review · Reviewer_ZPse · 2021-07-26

**Rating:** 4
**Confidence:** 3

**Summary:**

This paper design four quantitative indicators of attack failures (IoAFs) that might help to systemically automatically check the correctness of robustness evaluations. Besides, the authors also provide several mitigation methods for these IoAFs. According to these indicators (IoAFs) as well as corresponded mitigation methods, the authors show the effectiveness of indicators (IoAFs) applied on several existing defense approaches and also proposed mitigation methods.

**Limitations And Societal Impact:**

The authors listed their limitation and meanwhile proposed the alternative method to address it.

**Main Review:**

Pros:
1. This work propose quantitative indicators to point out reasons of attack failures.
2. The experiments show the effectiveness of proposed indicators of attack failures (IoAFs) and mitigation strategies.

Cons:
1. Based on [1], we know that IoAFs can work for the correctness of robustness evaluations. However, it only designs metrics for known factors in this paper. It's not surprising that IoAFs are effective. It would be better if the authors have some non-heuristic or theory-driven reasons to explain why this works.
2. The proposed mitigation strategies have already been used in several papers. For instances, $M_2$, $M_4$ and $M_5$ were used in [2, 3]. Therefore, it exists several works showing that the proposed mitigation strategies are helpful.
3. The defense methods included in the experiments were defeated in existing works [4,5].

Overall, quantifying the indicators of attack failure is a novel idea. But the concepts of these indicators have already been proposed before and similar scenario also happened the proposal of mitigation strategies. Besides, the defense methods in the experiments were all defeated in recent works. Namely, these posited mitigation strategies might already exist or at least are pretty common.

[1] Obfuscated Gradients Give a False Sense of Security: Circumventing Defenses to Adversarial Examples.

[2] Diversity Can Be Transferred: Output Diversification for White- and Black-box Attacks.

[3] Enhancing Gradient-based Attacks with Symbolic Intervals.

[4] On Adaptive Attacks to Adversarial Example Defenses.

[5] Towards Evaluating the Robustness of Neural Networks.




**Time Spent Reviewing:**

8

---

> ### Author Response · Authors · 2021-08-09
> **Reply to Reviewer ZPse**
>
> We thank the reviewer for their comments and for giving us the possibility of clarifying the scope of our work. Our indicators aim to quantify known failures in adversarial robustness evaluations and to provide a systematic protocol to detect and fix flawed evaluations. While we agree that common knowledge on such failures and mitigations already exists, even though scattered across different papers, no systematic tools for debugging attacks and defense evaluations are available. Therefore, the community keeps repeating the same known errors over and over when conducting adversarial evaluations.
>
> For example, among the defenses broken in [1], some were broken by adopting known mitigations, even though such defenses were published after those mitigations were already known and widely discussed. We believe that a debugging tool such as our indicators of failure may have prevented these overly-optimistic, flawed evaluations. We thus firmly believe that pairing our indicators with robustness evaluations would help significantly improve this process by avoiding common mistakes, and we are already working to include our indicators in existing adversarial libraries.
>
> Finally, while the reviewer is right when saying that we only applied known mitigations on already-broken defenses, we would like to remark again that this was done on purpose, to show that the indicators work to detect and characterize known failures. The goal of our work is to provide a framework to systematize the detection of known failures and the application of the corresponding mitigations. It is not the scope of this work to find novel causes of failures, novel mitigations, or break new defenses. Conversely, novel indicators should be developed if new failures are found in future work, similar to what happens in software development (e.g., security testing only detects known vulnerabilities - but, even if it is common knowledge, developers still need proper tools to find and patch them).
> We hope that our response clarifies the scope of this work better and that the reviewer may reconsider its novelty and potential impact under this perspective.
>
> [1] Florian Tramér, Nicholas Carlini, Wieland Brendel, & Aleksander Madry. (2020). On Adaptive Attacks to Adversarial Example Defenses.

---

> > ### Comment · Reviewer_ZPse · 2021-09-01
> > **Reply to authors**
> >
> > The authors clarified the contribution of their work in rebuttal and I would like to thank them for it. Unfortunately, my concern remains.
> >
> > I admit the contributions of using indicators to debug the attack failure and quantifying the indicators. However, concepts of indicators are lack novelty. Besides, it evaluates indicators by fewer defenses or attacks. Because these indicators are without the theoretical guarantee, it would be more convinced to show the effectiveness of indicators with more defenses or other attacks.

---

> > > ### Author Response · Authors · 2021-09-06
> > > **Acknowledgement**
> > >
> > > We respectfully disagree with the reviewer on the fact that the indicators lack novelty. To the best of our knowledge, no work has ever defined such quantitative measures of failure for gradient-based attacks, and we would be more than thankful to the reviewer if they can point out some references of work that defines quantitative indicators of failure for adversarial robustness evaluations (rather than only qualitative, generic recommendations). We will nevertheless keep working both to improve the indicators and to extend our experimental analysis, as the reviewer suggested.

---

### Author Response · Authors · 2021-08-09
**Global answer**

We thank all the reviewers for the time spent reviewing our work. We are pleased that the reviewers recognized some of the contributions we propose in this work.
In particular, most of the reviewers praised the originality and the clarity of our work, appreciating the complete reproducibility of our results through the attached code.
In the following, we address the general comments they raised before replying to each of them in detail.

**Indicators are not for comparing attacks or defenses.** The purpose of our paper is to help defense authors perform correct evaluations of their work. Our ideas are not designed to say which of Defense A or Defense B is better, or which of Attack 1 or Attack 2 is better. This is better captured by the final attack success rate, but only if the attacks are correctly executed against the defenses (which is exactly what our indicators should help check).

**Some indicators are trivial.** We create our indicators to be as simple as possible to detect the common failure modes of existing defense evaluations. If the indicators were complicated, then it would be tricky to use them to debug attack failures (because it would be hard to separate the case where the indicator has been applied incorrectly versus the attack actually having a problem). In other words, the simplicity of the proposed indicators is a design choice. Furthermore, even though they are simple, we find they are well suited to capture the failures reported in current papers. In fact, these trivial checks are not applied in most of the state-of-the-art evaluations that have been reported [1]. For example, we have found that famous attack libraries (like Foolbox and Cleverhans) are not performing a simple-yet-crucial check that we found thanks to our indicators.

**Evaluation on already-broken defenses.** The focus of our paper is to show that the indicators we have developed can predict flawed evaluations. As such, we need to have access to both the incorrectly performed evaluation and the correctly performed break. We therefore take defenses that are known to be broken, and attacks that are known to work on them, to demonstrate the utility and effectiveness of our indicators.

**Contribution and novelty.** We remark that the main contribution of our work is to provide debugging tools that help defense developers to better evaluate their strategies, and not to help break new defenses.

[1] Florian Tramér, Nicholas Carlini, Wieland Brendel, & Aleksander Madry. (2020). On Adaptive Attacks to Adversarial Example Defenses.

---

### Decision · Program_Chairs · 2021-09-27

**Decision:**

Reject

**Comment:**

All reviewers acknowledged the usefulness of the proposed indicators for debugging attack failures, but at the same time, there are major concerns being raised regarding overlapping contributions to known literature and insufficient experiments for demonstrating the value of the indicators. The authors' rebuttal did not address the main concerns of the majority of the reviewers.

Here is a summary of the main concerns. I hope the authors find the review comments and summary useful for preparing the future version.

1. The introduced indicators are useful in debugging attack algorithms, but many of them are common practices or overlap with known results in existing literature, limiting the contributions of this work
2. Some indicators overlap with existing literature on adaptive attacks, such as the NeurIPS 2020 paper "On Adaptive Attacks to Adversarial Example Defenses"
3. This paper mainly focused on 4 defenses (that were previously shown to be broken) while earlier works of similar fashion such as "Obfuscated Gradients Give a False Sense of Security: Circumventing Defenses to Adversarial Examples" and "On Adaptive Attacks to Adversarial Example Defenses" show defense failures on about 10 defenses.